# Retrotransposons and the Evolution of Genome Size in *Pisum*

**DOI:** 10.3390/biotech9040024

**Published:** 2020-11-26

**Authors:** T. H. Noel Ellis, Alexander V. Vershinin

**Affiliations:** 1John Innes Centre, Norwich Research Park, Colney Lane, Norwich NR4 7UH, UK; 2Institute of Molecular and Cellular Biology, Acad. Lavrentiev Ave. 8/2, 630090 Novosibirsk, Russia; avershin@mcb.nsc.ru; 3Department of Natural Sciences, Novosibirsk State University, Pirogova 2, 630090 Novosibirsk, Russia

**Keywords:** genome size, retrotransposons, pea, legumes

## Abstract

Here we investigate the plant population genetics of retrotransposon insertion sites in pea to find out whether genetic drift and the neutral theory of molecular evolution can account for their abundance in the pea genome. (1) We asked whether two contrasting types of pea LTR-containing retrotransposons have the frequency and age distributions consistent with the behavior of neutral alleles and whether these parameters can explain the rate of change of genome size in legumes. (2) We used the recently assembled v1a pea genome sequence to obtain data on LTR-LTR divergence from which their age can be estimated. We coupled these data to prior information on the distribution of insertion site alleles. (3) We found that the age and frequency distribution data are consistent with the neutral theory. (4) We concluded that demographic processes are the underlying cause of genome size variation in legumes.

## 1. Introduction

Variation in the size of nuclear genomes among organisms has been a long-standing area of interest [1,2].

Within the legumes (Leguminosae, or Fabaceae), genome sequences are now available for a broad diversity of Papilionoid (Faboid) taxa [3] and these show that legume genomes typically have ca. 37,000 ± 10,000 annotated genes, similar to that for angiosperms more widely [4]. Among diploid legume species, genome size ranges about 40-fold, from ca. 340 Mb in several *Trifolium* species to a little over 14,000 Mb in *Lathyrus vestitus* [5]. Genome size variation among legumes is in contrast to their relatively constant gene number. However, genome size in *Pisum* seems to be stable, despite underlying variation in the presence and absence of retrotransposon insertions [6,7]. The one exception to this stability is the approximately 10% larger genome size noted in *P. abyssinicum* and *P. fulvum* [6], which are notably distinct taxa [7] within the genus.

Much of the variation in diploid legume genome size is attributable to variation in the content of LTR (long terminal repeats) retrotransposons [8,9,10]. Retrotransposons replicate by a copy and paste mechanism [11] and so they have the potential to accumulate to a great extent in nuclear genomes. It was suggested that this behavior implies that genome size should increase irrevocably [12] unless mechanisms exist by which retrotransposons may be removed [13,14,15,16,17,18]. This process was discussed recently by Jedlicka et al. [19].

In this study, our aim is to investigate the properties of retrotransposon insertions in the *Pisum* genome in order to constrain population genetical models of their dynamics. This requires a description of the age and frequency distribution of retroelement insertions in order to put limits on population genetic parameters of the neutral theory [20]. The details of these models are described in Section 2 below. We selected two contrasting elements for this analysis. The first, *PDR1*, is a *Ty1/copia* superfamily retrotransposon present in about 200 copies per haploid genome, evenly distributed along all pea chromosomes as was shown genetically [21], by in situ hybridization (Vershinin, unpublished data) and as is clear from the available genome assembly [22]. *PDR1* is about 4 kb in length, and its LTRs, at 156 bp [23], are exceptionally short. The second, *Cyclops*, has the typical pol region of the *Ty3/gypsy* superfamily of retrotransposons and is present in about 5000 copies [24]. *Cyclops* elements are approximately 12 kb long, including very long LTRs of about 1500 bp.

Previous studies in *Pisum* reached two important conclusions about its retroelement content; the first is that allelic variation in the genus *Pisum* is very broadly distributed and “recombination, introgression, and segregation between pea inbred lineages is common, although this may be rare per plant generation” [7]. The second conclusion is that the average age of retrotransposon insertions is one to two Myr [25]. Now that a genome sequence of *Pisum* was been assembled [22], further study of divergence between LTRs of individual elements and a more complete understanding of their genomic location is possible. Here, we are interested in how treating retrotransposition as an analogue of neutral base substitution provides insight into the expected age and frequency distribution of retrotransposon insertions. In other words we are asking whether genetic drift alone can explain the variation in genome size in the *Viceae*.

## 2. Materials and Methods

### 2.1. Plant Material and the Selection of Accessions for Analysis

Accessions from the John Innes Pisum Germplasm are designated JI*x*, where *x* is a number [26]. The analysis of this collection was carried out by the SSAP (Sequence Specific Amplification Polymorphism) technique and the data obtained were used to generate a pairwise distance matrix of allelic differences [7]. Principal coordinate analysis was used to order the distance matrix of all pairwise differences, and reduce to one member, pairs or groups of accessions that shared 95% or more of the marker alleles. A selection of 44 accessions was made from these data after excluding those that were closely related. This eliminated one *P. sativum* accession (JI 188), two *P. sativum* ssp *transcaucasicum* accessions (JI 2547 & JI 196), and four *P. abyssinicum* accessions (JI 1556, JI 2385, JI 130 and JI 2); leaving the following accessions: *P. abyssinicum*: JI 225; *P. fulvum*: JI 224, JI 1006, JI 1010, JI1796; *P. elatius*: JI 64, JI 254, JI 261, JI 262, JI 199, JI 1074, JI 1092, JI 1093, JI 1096, JI 2201, JI 2055, and JI 1794 (sometimes called *P. humile*). The *P. sativum* accessions included JI 45 and JI 2546 (designated ssp *transcaucasicum*), JI 156, JI 185, JI 189, JI 281 (African landraces), the Asian landraces JI 85, JI 95, JI 102, JI 109, JI 181, JI 241,JI 804, JI 1346, JI 1428, JI 1854, JI 2545, JI 250 (sometimes called *P. jomardii*), JI 52, JI 201, JI 209, JI 284, JI 399, JI 1030, JI 1089, JI 1846, and JI 2713. All accessions are available from the John Innes *Pisum* germplasm collection [26].

### 2.2. Population Genetic Considerations

The effective population size is the number of individuals that would be needed to generate any given statistic of population genetics for the population, if it comprised a set of individuals that interbreed freely and at random, i.e., are in Hardy–Weinberg equilibrium.

#### 2.2.1. Allele Frequency Distribution

The presence or absence of retrotransposons at individual locations in the pea genome was observed by the SSAP technique [21]. We treated these data as genetic loci with two allelic states. The ancestral condition, which is the absence of an insertion, is called the unoccupied or empty site, and an evolutionarily derived allele, the occupied site, is defined by the insertion of a retrotransposon at this previously unoccupied site. The derived allele can suffer subsequent loss of the internal region (between the LTRs) by LTR–LTR recombination creating a solo LTR, or the deletion of the genomic region carrying the insertion. These events are not discussed further as they occur in a fraction of the individuals in the population that carry an occupied site allele, and their subsequent behavior would follow the same trajectory as the initial insertion allele.

Retrotransposon insertion creates a new allele with a frequency (*p*) for the occupied site and the frequency of the empty site becomes (1 − *p*); initially *p* = 1/2*N*, where *N* is the population size, and the factor of 2 is because the species is diploid. These values define the effective heterozygosity; *H_e_* = 2*p* (1 − *p*), which is the chance that two alleles chosen at random are different. Effective heterozygosity, for neutral alleles, is related to the population genetic parameters of effective population size (*N_e_*) and mutation rate (*μ*) [20,27,28,29]:(1)$$4N_{e}\mu =\;H_{e}/\left(1-\;H_{e}\right)\;;\;\;\;\;\;\mu =\;H_{e}/4N_{e}\left(1-\;H_{e}\right)$$

When 4*N_e_μ* is estimated from *p*, the lowest frequency for which an allele can be observed is the reciprocal of the number of individuals that were genotyped. As *p*→ 0, 4*N_e_μ*→ 2*p* determines the resolution of the observable values of 4*N_e_μ*.

Furthermore, the expected frequency distribution of the abundance of an allele Φ(*x*) is determined by the effective population size *N_e_* and the mutation rate *μ* [20,27,29] as follows:(2)$$\Phi \left(x\right)=4N_{e}\mu {\left(1-x\right)}^{4N_{e}\mu }/x$$

We used the average frequency of occupied sites to determine the expected frequency distribution using Equations (1) and (2). We then determined whether or not the observed data were a good fit to this expectation using χ^2^ test. For clarity, we used the term *ρ* for the retrotransposition rate to avoid confusion with single base substitution, *μ*.

The SSAP data are available in Appendix A.

#### 2.2.2. Age Distribution of Occupied Sites

LTR retrotransposons replicate by a copy and paste mechanism [11] where a transcript is initiated in the 5′ LTR and terminated in the 3′ LTR. Reverse transcription of this RNA and second strand synthesis generates a circularly permuted intermediate dsDNA where the LTR of this DNA is derived from one copy of the LTR sequence [30,31]. Upon insertion into the genome, the single LTR of the cDNA is replicated and defines the two ends of the element. Thus, at the time of insertion these two DNA sequences are derived from a single molecule and are therefore expected to be identical. Differences between these LTRs can accumulate due to mutation, and for this reason the comparison of the LTR sequences at an individual insertion site was used as a measure of the time since insertion, based on the assumption that these sequence differences arise by mutation at the same rate as silent substitution [32].

Using the pea v1a genome sequence [22] and prior data [7], we compared the age of an insertion estimated from LTR–LTR sequence divergence to the expected age of a neutral allele in a population, as determined by population genetic parameters. Kimura and Ohta [33] derived formulas for the age of neutral alleles that first achieve a given frequency in a population:(3)$${\bar{t}}_{x}\left(0\right)=4N_{e}\left\{\frac{1-x}{x}\ln\left(1-x\right)+1\right\}$$
where *x* is the frequency of a neutral allele in the population after an average of $\bar{t}$ generations, having started at a very low frequency (1/2*N*), which can be considered to be effectively 0. Note that age is independent of the retrotransposition rate as it describes the fate of an allele once it has been formed. Kimura and Ohta [33] showed that the average or expected age, *E*(*age*), of a neutral allele is a function of effective population size *N_e_* and the current frequency of that allele *x*, such that:(4)$$E\left(age\right)=\;-4N_{e}\left[x/\left(1-x\right)\right]\ln\left(x\right)$$

As *x* ranges between 0 and 1, the term [*x*/(1 − *x*)]ln(*x*) ranges between 0 and −1, which means that *E*(*age*) ≤4*N_e_.* We used estimates of *N_e_* obtained from the allele frequency distribution (above) to determine whether the observed (from LTR–LTR divergence) and expected (from Equation (4)) ages of retrotransposon insertion sites were compatible.

## 3. Results

### 3.1. Retrotransposons in Legumes

The taxonomic distribution of legume genome size (Figure 1) shows that the largest genomes occur within the Viceae (Fabeae) tribe, which includes *Pisum*, *Lathyrus*, *Vicia*, and *Lens.* The Viceae genomes are not uniformly large, but also contain species with genomes of a size more typical for legumes generally. The distribution of genome sizes within the Viceae is consistent with an evolutionary history of both increase and decrease in genome size (Appendix B, Figure A1).

The distribution of genome sizes in the Vicieae suggests that evolutionary change in diploid genome size occurred within 5 My (Figure 1 and Figure A1) and is therefore rapid, which is consistent with the differences being due to differential accumulation of retrotransposons.

### 3.2. Allele Frequency Distribution

Jing et al. [25] compared the observed and expected frequency distribution of insertion site alleles for the *Ty1/copia* element *PDR1* and found it a good fit to the expectation from the neutral theory. Here, we undertook the same analysis for the more abundant *Ty3/gypsy* element *Cyclops* [24] using the data from [7] in a selection of 44 pea accessions that represent the diversity of *Pisum* and that does not include multiple closely related accessions (see Section 2). The *PDR1* data for this subset of 44 accessions is compared to the frequency distribution of *Cyclops* insertion alleles in Figure 2.

Figure 2 shows that, for both plots, there are fewer alleles with a frequency in the range 0–0.04 than is expected from the neutral theory. Presumably, this is because frequencies less than 1/44 cannot be observed. There is also an excess for the ‘fixed’ class (allele frequency = 44/44), where all accessions carry the occupied site allele. This observed fixed class also includes alleles with a frequency greater than 44/45. Hence, this frequency class is expected to be overrepresented. That is, we cannot distinguish between insertion sites in all individuals in the genus from insertion sites present in just these 44 accessions.

The area under the curve corresponding to Φ(*x*) has to be estimated numerically, because the function has an improper integral; the area under the tails of the curve cannot be determined.

The occupied sites, which are present in only one accession, are distributed widely, for *PDR1* there are 25 of these, while for *Cyclops* there are 21. Of these, 4 are in the single *P. abyssinicum* accession and 9 in the 4 *P. fulvum* accessions, consistent with the differentiation of these taxa.

A χ^2^ test for the observed vs. expected number of alleles in each frequency class, other than the two extremes, shows which observed values are significantly different from expectation. For this test, all frequency classes with an expectation less than or equal to 5 were combined into a single group. For both retrotransposons, a single class (ringed in Figure 2) had a significant value, χ^2^ = 4.84 & 5.43, *p* = 0.0278, and 0.020 for *PDR1* and *Cyclops*, respectively. For *PDR1* occupied site allele frequencies ≥ 0.56, the expected number was equal to or less than 5, so these were treated as a single class, χ^2^ = 0.001, *p* = 0.98. For *Cyclops* occupied site allele frequencies ≥0.76, the expected number was equal to or less than 5, so these were treated as a single class, χ^2^ = 0.86, *p* = 0.35. These data suggest that with the exception of the fixed alleles and the lowest frequency class, the data are an excellent fit to the prediction of the neutral theory. We know from the discussion above, that the fixed alleles and lowest frequency class do not have a properly defined expectation. If we accept the interpretation that the data are a good fit to the neutral theory, then Equation (2) suggests that only ca. ¼ of occupied site alleles expected to be found with a frequency less than 0.04 were detected in this sample of accessions. For both retrotransposons, this is about 1/3 of the total number of occupied sites detected, implying that we have detected about 75% of the number of insertion sites in *Pisum* that could be detected in a sample of this size. The expected abundance of alleles with a very low frequency is arbitrarily large, implying that a very large number of insertions are extremely rare and are very quickly lost from the population.

With the exception of the extreme values discussed above, the neutral theory appears to give an adequate description of the frequency distribution of occupied site alleles of retrotransposons in *Pisum.* The estimated values of 4*N_e_ρ* are remarkably similar for *PDR1* and *Cyclops*, two very different elements. This is not expected and implies that the survival rate of new insertions in the population is similar. We therefore asked whether the neutral theory can also explain the age distribution of retrotransposon insertions.

### 3.3. The Distribution of Cyclops Elements in the Cameor Genome Assembly

*Ty3-gypsy* elements are often described as being clustered in pericentric regions, as for example in *Arachis* [35]. Using the theory of runs [36] to examine the location of *Cyclops* elements in the pea cv. Cameor genome [22] provided no evidence for their having a non-random distribution at the scale of 100 kb blocks (Appendix D). The low recombination pericentric regions occupy ca. 720 Mb or roughly 18% of the 3.92 Gb assembly. Given the random distribution of *Cyclops* elements, we expect ca. 18% to lie within these low recombination regions. This should correspond to about 60 of the insertion sites assayed in the genetic diversity study.

### 3.4. Occupied Site Allele Age Distribution

*Cyclops* LTR sequences in the Cameor genome that were in the same orientation and separated by 8 to 10 kb were identified as candidates for pairs flanking a single element. This selection was further refined by removing sequences where more than two alignments with the LTR were found (Appendix D). This left 390 LTR sequences with the appropriate spacing and orientation. Neighbor Joining trees of these sequences were generated to test whether the adjacent LTRs were each other’s most similar sequences. This further stringent filtering step left a list of 49 LTR pairs that had the expected characteristics from a single insertion event (Appendix D). These paired LTRs were compared to each other using BLASTn, noting the alignment length and the number of mismatches (gap openings were ignored), to determine the number of substitutions between LTR pairs.

Twenty-five LTR pairs of the *PDR1* retroelement present in the Cameor genome were identified and filtered in a similar way, and the number of pairwise differences was determined. These data were compared to the 49 LTR pairs previously described by Jing et al. [25], as presented in Figure 3 and Table 1.

These estimates of sequence divergence are not significantly different from one another, or from the data of Jing et al. [25].

## 4. Discussion

We have investigated the age and location of two retrotransposons in pea genomes. *PDR1* is a *Ty1/copia* class element present in about 200 copies per genome, while *Cyclops* is a *Ty3/gyp*sy element present in about 5000 copies [23,24]. The insertion sites of these two contrasting types of retrotransposon have a similar age and frequency distribution in *Pisum*, therefore it seems plausible that common factors have shaped these features of the elements. Inevitably, they have shared a similar population biology of their host plant, which is one obvious factor in common.

### 4.1. Nucleotide Diversity and Effective Population Size

Jing et al. [37] estimated the nucleotide diversity (*π* = 4*N_e_μ*) among 39 genes in 46 *Pisum* accessions as 0.011 ± 0.007. Estimates of *π* are also available from Sulima et al. [38] based on three genes among 110 accessions and from the 30 sequences derived from 25 genes among 100 accessions analyzed by Carpenter et al. [39]; these are 0.019 ± 0.003 and 0.006 ± 0.005 (mean ± SD), respectively. Kreplak et al. [22] estimated the nucleotide diversity of *Pisum* as ca. 8 × 10^−4^, which is about an order of magnitude lower than in the other three studies. Estimates of nucleotide diversity depend on the range of accessions analyzed, and the first three data sets were designed to capture the diversity of *Pisum* as a whole, while Kreplak et al. [22] were primarily concerned with the sequence of the cultigen Cameor in the context of cultivated pea and its relatives; accordingly, this set was dominated by cultivated forms. These accessions included 16 cultivars, 15 landraces, 2 *P. abyssinicum*, and 10 wild accessions. This difference in the representation of wild accessions, which carry the bulk of the diversity of *Pisum*, is consistent with the lower estimate of *π* in Kreplak et al. [22]. An estimate of 4*N_e_μ* for *Pisum* as a whole in the range 0.005 to 0.01 is compatible with all these previous data. If we take the mutation rate as ca. 10^−8^, then the estimate of *N_e_* is ca. 3–4 × 10^4^.

### 4.2. Age Distribution of LTR Pairs and Effective Population Size

A critique of the LTR–LTR comparison method for dating the age of retrotransposon insertions was made by Jedlicka et al. [19], who claimed that biases exist, some attributed to conversion events, such that longer LTR pairs were more similar to each other than shorter LTR pairs. The authors also commented that this phenomenon was partly reproduced in data simulation (although the reason for this was not clear). It should be noted that LTR length is not independent of retrotransposon family and different retrotransposons may have different genomic locations that may contribute to differences in recombination and/or gene conversion rate. Potential gene conversion events were identified by comparing the “ratio of solo LTR/FL”—presumably comparing the sequence of solo LTRs with that of the paired LTRs of intact elements. This method assumes that the sequence diversity of solo LTRs and the LTRs of a given intact element is the same, which may not be the case because of subfamily structure within retroelements [40].

Furthermore, Jedlicka et al. [19] noted that for approximately a quarter of nested insertions, the targeted element appeared to be younger than the element that was subsequently inserted. There are several mechanisms by which this may occur, but the observation highlights the need for caution and emphasizes the possibility that recombination-like processes may lead to an underestimate of the divergence between LTR pairs. Nevertheless these authors note that, for a wide range of species, most estimates of the mean age of LTR retroelement insertions are in the range of 1–3 Myr, which is consistent with the estimates obtained here.

From Equation (4), the expected age of any allele with a frequency *x* = 0.28 (the average of the frequencies for both *PDR1* and *Cyclops*) is ca. 60,000 to 120,000 years. This means that the measured age of retrotransposon insertions (1–3 Myr) is very much greater than expected, but can be understood as follows (see also Appendix E). Retrotransposon insertions, which carry a sequence difference between the LTRs are necessarily derived alleles; they must have occurred in a pre-existing insertion. The expected age in Equation (4) corresponds to the length of time until the insertion allele *first* reaches the frequency *x*, not the average age of an allele of this frequency. An insertion in which there is one difference between the LTRs arose from an insertion allele in which the LTRs were identical. The derived allele was therefore at the frequency 1/2N when the original insertion event occurred, and again 1/2N when the mutation defining the derived allele occurred. The number of times an insertion allele has visited the frequency 1/2N is therefore at least equal to the number of differences between the LTRs. Each time this occurs, the probability that the allele will be lost by chance alone is high, thus we do not expect a large number of SNP variants per insertion site, nor do we expect such variants to exist at an appreciable frequency in *Pisum* as a whole.

With a nucleotide substitution rate of ca. 10^−8^, for a retroelement with LTR length 0.1 to 1 kb a single base change will on average occur within about 10^5^ to 10^6^ years. If this variant reaches a moderate frequency in the population, then a further period of about 10^4^ to 10^5^ years will have elapsed. Thus, the estimated age of retrotransposons of the order of 1–2 million years is consistent with the mutation rate in the LTRs and the population dynamics that permit only a few of these derived alleles to achieve a moderate frequency.

### 4.3. Gain and Loss

The similar estimates of 4*N_e_ρ* (Figure 2, i.e., the equivalent of *π* for retrotransposon insertions) above suggest that the long-term transposition rate *ρ* is very similar for *PDR1* and *Cyclops*, and is about 1.5 × 10^−7^. The similarity of these two values of *ρ* may simply reflect a long-term average, with transposition rate varying between the elements from time to time. It is necessarily the case that *ρ* is the transposition rate for insertion sites that survive in the population, which is not necessarily the same as the rate at which retrotransposition occurs in a given individual.

Our study suggests that there is little remarkable about the age and frequency distribution of retrotransposon insertion site alleles in pea, yet we know that pea and its relatives in the Viceae present a diversity of genome sizes. Many of these species have genome sizes larger than pea, and others with smaller genomes. The size difference in these genomes seems to be accounted for by differential accumulation of retrotransposons [8,10], and the taxonomic distribution of genome size suggests that both gain and loss has occurred (Appendix B, Figure A1). The overall abundance of retrotransposons could change if their transposition rate changed coordinately. However, it seems more likely that this genome-wide property is a consequence of the period of time for which they remain in the genome, and this is determined solely by effective population size. Thus, we propose that historical (evolutionary) changes in effective population size are the main reason for the diversity of genome size in the Viceae. An increase in effective population size, all other things being equal, would lead to an accumulation of (polymorphic) retrotransposon insertions and hence an increase in average genome size. On the other hand, a reduction in effective population size would reduce allelic diversity by facilitating the loss of alleles. Reduction in effective population size would not necessarily reduce genome size, but would replace the mean genome size with the mean genome size of a sub-population. However, with a reduced effective population size, the number of polymorphic alleles that could accumulate would be reduced, so the effectiveness of retrotransposition to increase average genome size would be reduced.

In general, recombination rate per kb is negatively correlated with genome size [2] and, through hitchhiking effects, recombination rate influences the effective population size [2,41] such that the effective population size is increased in regions of higher recombination rate. With higher effective population size, the average age of alleles is increased (Equation (4) above). So regions of high recombination rate will include polymorphic retrotransposon insertions, which would be fixed (as either the empty or occupied site) more rapidly in regions of low recombination or lower effective population size.

Bertioli et al. [35] showed that the *A. ipaensis* genome is 10%–20% larger than that of *A. duranensis* with more frequent duplications and a higher transposon content. Several of the corresponding chromosomes in these genomes differ by having a large distal inversion so that the telomeric region of one is closer to the centromere in the other. The alignment of the pseudomolecules shows that the physical distance between matched sequences is longer in the species where these are nearer the centromere and shorter where these are nearer the telomere. This effect is continuous and gradual, as revealed by an arc in dot-plots of homeologous chromosomes [35], and is associated with a difference in transposon abundance; the extra transposons accounting for the increased length. These inversions have moved sequences from a region of high recombination (closer to the telomere) to a region of low recombination (closer to the centromere) and the consequence of this change is seen in the repetitive sequence content of these regions of the genome.

The occupied site allele for a retrotransposon is initially rare, so these are usually lost by genetic drift, but, by chance, a few may become relatively abundant in the population. A change in effective population size can have a systematic effect, as in the example from *Arachis* [34]. In pea, the recombination rate per kb is low with respect to its close relatives with smaller genomes. This is simply because chromosome arms, irrespective of size, typically have 1 or 2 crossovers; chromosomes require at least one crossover for proper disjunction. As recombination rate per kb simply describes the situation, it cannot be taken as an explanation for the increased retrotransposon content in pea (and many other members of the Viceae) compared to other legumes. However, if effective population size in a small genome ancestor of pea increased, then a longer time would have to elapse before the loss of insertion alleles and this effect, therefore, may have led to an abundance of polymorphic retrotransposon insertions as is seen in extant pea lineages.

### 4.4. Comparison with Other Studies

Our general conclusion from these observations is that the age and frequency distribution of *PDR1* and *Cyclops* retrotransposons in pea can be accounted for according to the Neutral Theory. In other words, their age and abundance are dominated by demographic processes. We infer that these processes would act on the genome as a whole, although they would be modulated somewhat by local genomic effects on effective population size. For this reason, we would expect to see coordinated behavior of retrotransposons and in consequence their age distribution of insertions would be similar, reflecting these events. Note that the age distribution of insertions is entirely distinct from the age distribution of an element; an element may be much older than its individual incarnations, which successively occupy sites that are fleetingly present in the population.

Jedlicka et al. [19] used two methods to estimate the frequency distribution of the ages of insertions of 22 retroelement families in 15 diverse plant taxa. Both methods show broad similarity in the age distribution of insertion sites of different elements within each species, but with clear differences between species. In their ‘complex’ approach [19], attempting to account for conversion events, there is for example a marked bimodal age distribution for several elements in tomato. This type of pattern would be expected in species which have undergone changes in effective population size within the period of time that these insertion sites have survived.

A notable feature of pea is that it is predominantly self-pollinating. This led us to suspect that it should have a small effective population size as compared to the expectation under outcrossing. In turn, this would lead us to expect a relatively small genome size. However, as Vershinin et al. [7] noted, *Pisum* diversity is marked by recombination, introgression, and segregation. Presumably, this reflects outcrossing between stands of relatively homogeneous and homozygous individuals (Appendix E). The persistence of these stands should be assisted by self-fertility, and their persistence is required for successful outcrossing.

Macas et al. [9] have shown that *Ogre* elements represent by far the greatest bulk of LTR retrotransposons in the Viceae and that variation in their copy number is most strongly correlated with genome size within this tribe. Furthermore, although *Ogre* elements are present in other eudicots, including the Trifoleae, sister to the Viceae, it is only within the Viceae that they have reached such a high fraction of the genome [9]. These authors showed that *Ogre* elements are the main drivers of genome size variation in this tribe, while recognizing that “contrasting population sizes and different ecological and mating strategies” are likely to be significant forces shaping the retroelement composition of plant genomes. Here, we argued that effective population size and transposition rate together define these dynamics. The amplification and diversification of elements is represented by variation in transposition rate, but the dynamics of their accumulation or elimination needs to be understood in terms of population genetical history.

## 5. Conclusions

We propose that the uniformity of genome size in *Pisum* reflects the randomization of insertion alleles throughout the genus, rather than their fixation. Treating retrotransposon insertions as effectively neutral alleles can explain their age and frequency distribution in *Pisum*. If the elements we analyzed are representative of all pea retrotransposons, we can conclude that genetic drift alone can explain the variation in genome size in the *Viceae*. This further suggests that a large effective population size, which would maintain a high level of insertion site polymorphism, is the underlying cause of the large genome size in pea.

## Figures and Tables

**Figure 1 biotech-09-00024-f001:**
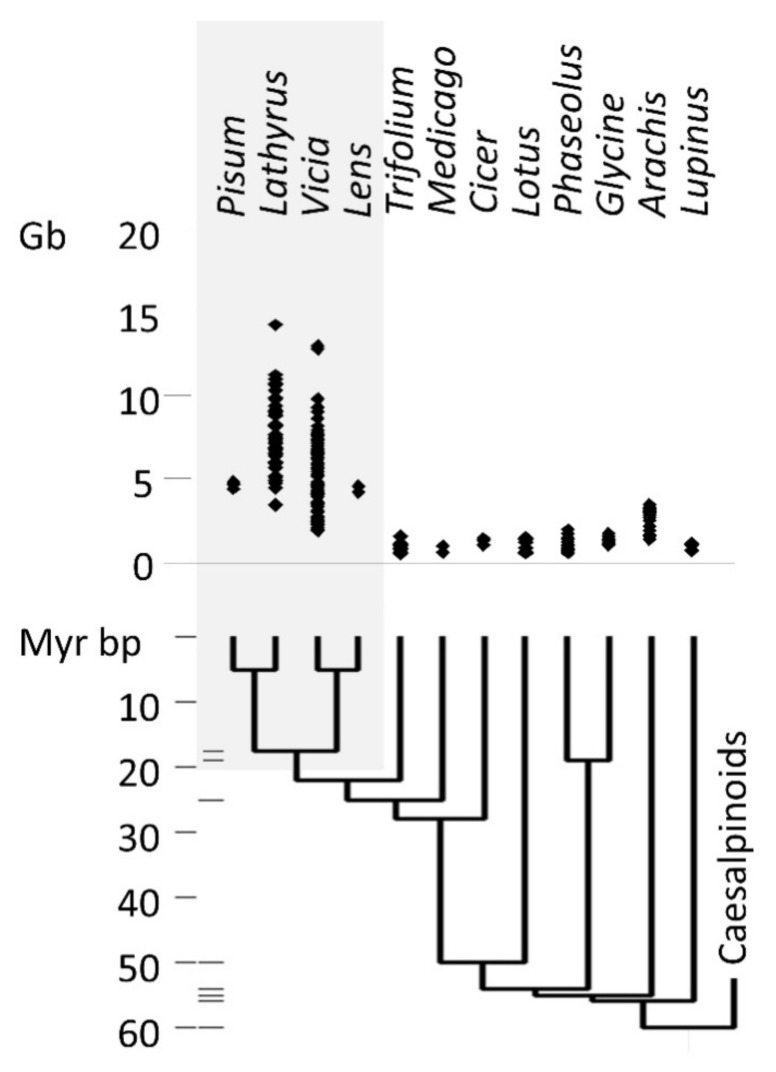
Diploid genome sizes in legumes. Genome sizes are from the plant C-value database [5]. The phylogenetic tree and the dates for divergences are from [34]. Grey shading indicates taxa in the tribe Viceae; horizontal bars to the right of the 10 Myr time scale represent splits supported by fossil evidence [34].

**Figure 2 biotech-09-00024-f002:**
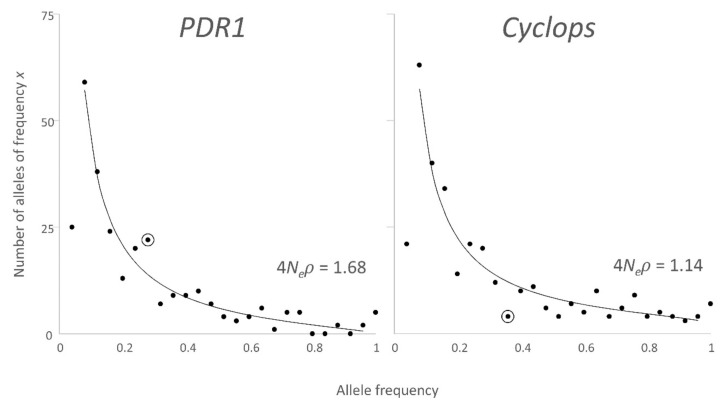
Frequency distribution of insertion alleles. The occurrence of occupied sites was observed for 329 *Cyclops* and 281 *PDR1* SSAP markers within a set of 44 accessions representing *Pisum* diversity [7]. The frequency of these occupied site allele is given on the x axis, binned in groups of 0.04 (0–0.04, 0.04–0.08, … 0.96–1). The y axis is the number of alleles within the frequency class on the x axis. The black line is the fit of Φ(*x*) with the minimum total χ^2^. The curve for Φ(*x*) has 4*N_e_ρ* as 1.68 and 1.14 for *PDR1* and *Cyclops*, respectively (Appendix A, Appendix C). The χ^2^ test showed that the highlighted values (ringed) differ from expectation at 5%, but not 1%, level. Note that in Equation (2) where *x* = 0, Φ(*x*) is unbounded.

**Figure 3 biotech-09-00024-f003:**
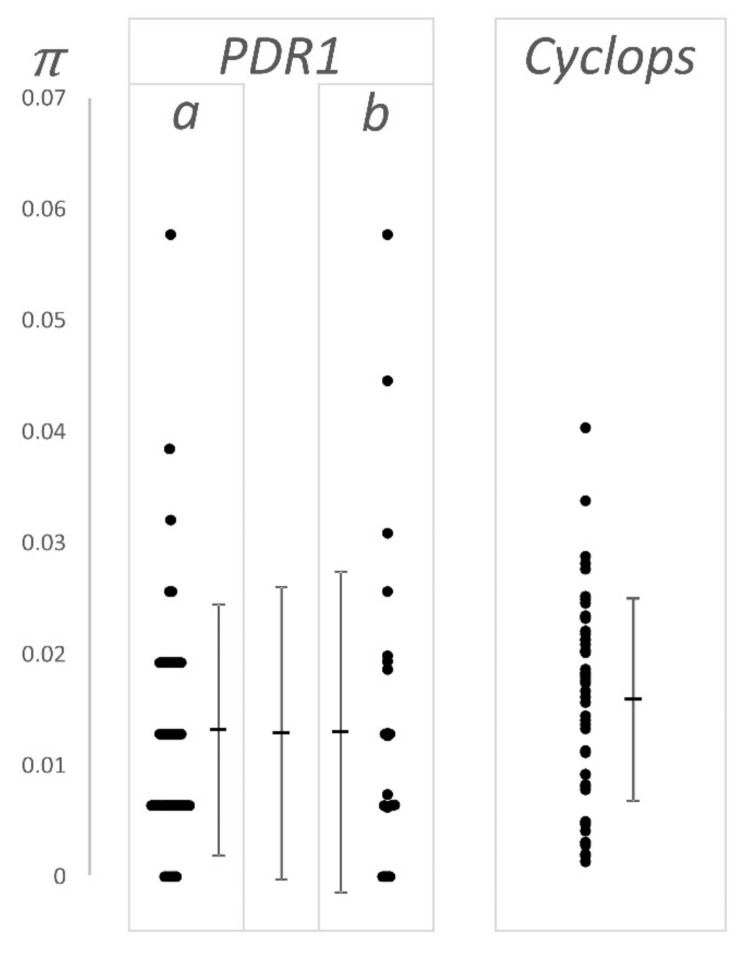
LTR–LTR divergence. The y axis, π (= 4 *N_e_μ*), is the fraction of single nucleotide substitutions observed over the length of the compared sequences. This ignores the variation due to indels, but includes their length. The data for *PDR1* are (**a**) taken from [25], (**b**) derived from the Cameor genome sequence [22] as is the data for *Cyclops*. All individual values are plotted (side by side when they have the same value) and the mean and standard deviation of the values are also plotted (see also Table 1). For *PRD1*, the central mean and standard deviation is for the combined data set. The bunching of values, giving a ladder-like appearance to the *PDR1* data is because this LTR is short; the observable values of π increment by the reciprocal of the LTR length, e.g., the first ‘rung’ of points above zero for *PDR1* is at 1/156 (ca. 0.006).

**Table 1 biotech-09-00024-t001:** LTR pair divergence.

Element	μ ± SD, *n* ^1^	Estimated Age ^2^	Source
*PDR1*	0.013 ± 0.011, 49	1.89 ± 0.33	[25]
*PDR1*	0.013 ± 0.014, 25	1.86 ± 0.55	This work, [22]
*PDR1*	0.013 ± 0.013, 74	1.88 ± 0.30	combined
*Cyclops*	0.016 ± 0.009, 49	2.20 ± 0.30	This work, [22]

^1^ Fraction of pairwise substitutions, n number of LTR pairs. ^2^ Myr (μ ± SEM).

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
