# Peer review of "Retrotransposons and the Evolution of Genome Size in Pisum"

_biotech, 2020, doi:10.3390/biotech9040024_

Round 1

Reviewer 1 Report

The manuscript is dedicated to the study of retrotransposon insertions in the Fabaceae genomes. The genomes of legumes vary greatly in their size, which is obviously connected with insertions of retrotransposons. The idea that the increasing of LTRs is connected with effective population size, seems very interesting and important. The only disadvantage of the manuscript is the complexity of the language. I would recommend to add some clarifications and explanations to the introduction and results section. For example, some parts of M&M section might be moved to the introduction.

Author Response

Response to reviewers' comments

We thank the reviewers for their helpful and constructive comments. Our response to these comments is given below where the reviewers' comments are in italic font immediately before our response.

The revised text is uploaded in track changes mode with respect to the original version and also as a version where all changes have been accepted because the changes to the text are rather complicated at some places.

We did not use track changes for the replacement of figures (Figure 2 - see response to Referee 1) and also the former Figure B3 (now Figure C3) where the axes were not labelled.

We hope that these alterations to the text are satisfactory.

Reviewer 1

The manuscript is dedicated to the study of retrotransposon insertions in the Fabaceae genomes. The genomes of legumes vary greatly in their size, which is obviously connected with insertions of retrotransposons.

This is in fact correct, but variation in diploid genome size could also be due to the accumulation of tandemly repeated DNAs. i.e. satellite DNAs as is the case in some mammals notably in the genus Dipodemus where satellite DNAs represent more than half the nuclear genome of Dipodemus ordii (Hatch and Mazrimas 1974 NAR 1: 559 - 576). However, Macas et al (2015), as cited in the text, showed that in this group of legumes the bulk of the variation in genome size is due to differences in the abundance of retrotransposons, notably the Ogre elements.

The idea that the increasing of LTRs is connected with effective population size, seems very interesting and important.

This is not exactly what we argued. Retrotransposons cannot accumulate without retrotransposition, so this must be a major reason for their accumulation. Rather we were interested in why they do not accumulate inexorably as Bennetzen and Kellogg (1997) discussed and several authors (as cited in the manuscript) suggested there should be some "DNA removal" mechanism as in the "increase decrease" model of Vitte and Panaud (2005). The main point of this paper is that there is no need to propose such a mechanism as population dynamics of the organism will cause changes in allele frequency and we showed that the magnitude these changes are large enough to account for the variation in genome size among diploid legumes and are consistent with age estimates of the individual insertions..

The only disadvantage of the manuscript is the complexity of the language. I would recommend to add some clarifications and explanations to the introduction and results section.

We recognize that this was a major problem with our text, so we have extensively re-written the text to make it clearer and easier to follow. We have simplified Figure 2 and the description of this figure. Furthermore we have added a new appendix to explain how the curve fitting was done and to show how different values of the main population genetic parameter change the shape of the curve. In making this change two errors were noticed in the data set: One Cyclops insertion site was absent from all 44 accessions (it was present in one of the accessions that was excluded from the analysis); This was removed bit doing so changed the number of insertion sites examined and so had a small effect on some of the numerical values. Similarly one of the PDR1 insertion sites had a score missing for one accession and this should have been an empty site. Correcting this error had a small effect on the numerical values and this has been corrected. Neither of these corrections are of a magnitude large enough to alter the conclusions of the paper and the data is available as supplementary data.

For example, some parts of M&M section might be moved to the introduction.

We felt that the flow of the text in the introduction should be maintained for the non-specialist reader, so adding details from the materials and methods, notably the population genetics section, would not improve the readability of the text. However, we have added additional reference to the materials and methods section to make it clear to the reader that further details are available if needed. We have also revised the text of this section carefully to improve its readability.

Reviewer 2 Report

An interesting study with good results. The evolutionary trends shown here by the rates of transposition were in accordance with that of genome itself. In other organisms happen similarly. Could be improved contrasting with analysis of gene sequences.

Author Response

Response to reviewers' comments

We thank the reviewers for their helpful and constructive comments. Our response to these comments is given below where the reviewers' comments are in italic font immediately before our response.

The revised text is uploaded in track changes mode with respect to the original version and also as a version where all changes have been accepted because the changes to the text are rather complicated at some places.

We did not use track changes for the replacement of figures (Figure 2 - see response to Referee 1) and also the former Figure B3 (now Figure C3) where the axes were not labelled.

We hope that these alterations to the text are satisfactory.

Reviewer 2

An interesting study with good results. The evolutionary trends shown here by the rates of transposition were in accordance with that of genome itself. In other organisms happen similarly.

We thank this reviewer for these comments.

Could be improved contrasting with analysis of gene sequence.

We agree that the behavior of gene sequences in the light of the Neutral Theory is of considerable interest, and referred to the work of Jing et al. (2007) on this subject. We also agree that it would be of considerable interest to compare the dynamics of allele age and frequency distribution for genic sequences. However, we think such a study would be rather large and would not really address the underlying question of whether genetic drift could account for genome size variation in the legumes. A further complication is that some genes will be subject to purifying selection, and some SNP variation will be in linkage disequilibrium with sequences under selective constraint. Yet other genes will be subject to diversifying selection and in such sequences linkage disequilibrium will again confound analysis (but in a different direction). We fully agree that this is an interesting subject in its own right and hopefully these issues can be addressed when genome wide sequence data is available from more Pisum taxa.